# Anticipating Performativity by Predicting from Predictions

**Celestine Mendler-Dünner**
Max Planck Institute for Intelligent Systems, Tübingen
cmendler@tuebingen.mpg.de

**Frances Ding**
Univerity of California, Berkeley
frances@berkeley.edu

**Yixin Wang**
University of Michigan
yixinw@umich.edu

## Abstract

Predictions about people, such as their expected educational achievement or their credit risk, can be performative and shape the outcome that they are designed to predict. Understanding the causal effect of predictions on the eventual outcomes is crucial for foreseeing the implications of future predictive models and selecting which models to deploy. However, this causal estimation task poses unique challenges: model predictions are usually deterministic functions of input features and highly correlated with outcomes. This can make the causal effect of predictions on outcomes impossible to disentangle from the direct effect of the covariates. We study this problem through the lens of causal identifiability. Despite the hardness of this problem in full generality, we highlight three natural scenarios where the causal effect of predictions can be identified from observational data: randomization in predictions, overparameterization of the predictive model deployed during data collection, and discrete prediction outputs. Empirically we show that given our identifiability conditions hold, standard variants of supervised learning that predict from predictions by treating the prediction as an input feature can find transferable functional relationships that allow for conclusions about newly deployed predictive models. These positive results fundamentally rely on *model predictions being recorded during data collection*, bringing forward the importance of rethinking standard data collection practices to enable progress towards a better understanding of social outcomes and performative feedback loops.

## 1 Introduction

Predictions can impact sentiments, alter expectations, inform actions, and thus change the course of events. Through their influence on people, predictions have the potential to change the regularities in the population they seek to describe and understand. This insight underlies the theories of performativity [38] and reflexivity [62] that play an important role in modern economics and finance. Recently, Perdomo et al. [51] pointed out that the social theory of performativity has important implications for machine learning theory and practice. Prevailing approaches to supervised learning assume that features $X$ and labels $Y$ are sampled jointly from a fixed underlying data distribution that is unaffected by attempts to predict $Y$ from $X$. Performativity questions this assumption and suggests that the deployment of a predictive model can disrupt the relationship between $X$ and $Y$. Hence, changes to the predictive model can induce shifts in the data distribution. For example, consider a lender with a predictive model for risk of default – performativity could arise if individuals who are predicted as likely to default are given higher interest loans, which make default even more likely [41], akin to a self-fulfilling prophecy. In turn, a different predictive model that predicts smaller risk and suggests offering more low-interest loans could cause some individuals who previously looked risky

to be able to pay the loans back, which would appear as a shift in the relationship between features $X$ and loan repayment outcomes $Y$. This performative nature of predictions poses a challenge to using historical data to predict the outcomes that will arise under the deployment of future models.

## 1.1 Our work

In this work, we aim to understand under what conditions observational data is sufficient to identify the performative effects of predictions. Only when causal identifiability is established can we rely on data-driven strategies to anticipate performativity and reason about the downstream consequences of deploying new models. Towards this goal, we focus on a subclass of performative prediction problems in this paper where performative effects of predictions solely surface as a shift in the outcome variable, and the distribution over covariates $X$ is unaffected by the prediction $\hat{Y}$. Our goal is to identify the expected counterfactual outcome

$$\mathcal{M}_Y(x, \hat{y}) \triangleq \mathbb{E}[Y|X = x, \mathrm{do}(\hat{Y} = \hat{y})].$$

Understanding the causal mechanism $\mathcal{M}_Y$ is crucial for model evaluation, as well as model optimization. In particular, it allows for offline evaluation of the potential outcome $Y$ of an individual $X$ subject to a predictive model $f_{\mathrm{new}}$ with the prediction $\hat{Y} = f_{\mathrm{new}}(X)$ before actually deploying it.

**The need for observing predictions.** We start by illustrating the hardness of performativity-agnostic learning by relating performative prediction to a concept shift problem. Using the specifics of the performative shift, we establish a lower bound on the extrapolation error of predicting $Y$ from $X$ under the deployment of a new model $f_{\mathrm{new}}$ that is different from the model $f_{\mathrm{train}}$ deployed during data collection. In particular, the extrapolation error grows with the distance between the prediction functions of the two models and the strength of performativity. This lower bound on the extrapolation error demonstrates the necessity to take performativity into account for reliably predicting $Y$.

**Predicting from predictions.** We then explore the feasibility of learning performative effects when the training data recorded the predictions and training data samples $(X, Y, \hat{Y})$ are available. As an identification strategy for learning $\mathcal{M}_Y$, we focus on building a meta machine learning model that predicts $Y$ for an individual with features $X$, subjected to a prediction $\hat{Y}$. We term this data-driven strategy *predicting from predictions*; it treats the predictions as an input to the meta machine learning model. The meta model seeks to answer "what would the outcome be if we were to deploy a different prediction model?" Crucially, this "what if" question is causal in nature; it aims to understand the potential outcome under an intervention which is different from merely estimating the outcome variable in previously seen data. Whether such a transferable model is learnable depends on whether the training data provides causal *identifiability* [49] Only after causal identifiability is established can we rely on observational data to select and design optimal prediction models under performativity.

**Establishing identifiability.** For our main technical results, we first show that, in general, observing $\hat{Y}$ is *not* sufficient for identifying the causal effects of predictions. In particular, if the training data was collected under the deployment of a deterministic prediction function, the mechanism $\mathcal{M}_Y$ can not be uniquely identified. The reason is a lack of coverage in the training data as $X$ and $\hat{Y}$ are deterministically bound. Next, we establish several conditions under which observing $\hat{Y}$ is sufficient for identifying $\mathcal{M}_Y$. The first condition exploits the presence of randomness in the prediction. This randomness could be purposely built into the prediction for individual fairness, differential privacy, or other considerations. The second condition exploits the property that predictive models are often over-parameterized, which leads to incongruence in functional complexity between different causal paths, enabling the effects of predictions to be separated from other variables' effects. The third condition takes advantage of discreteness in predictions such that performative effects can be disentangled from the continuous relationship between covariates and outcomes. Together, these results reveal that particularities of the performative prediction problem can enable us to recover the causal effect of predictions from observational data. In particular, we show that, under these conditions, standard supervised learning techniques can be used to find these transferable functional relationships by treating predictions as model inputs. Empirically, we demonstrate that supervised learning succeeds in finding $\mathcal{M}_Y$ even in finite samples.

We conclude with a discussion of limitations and extensions of our work, pointing out potential violations of the modeling assumptions underlying our causal analysis and proposing directions for future work.

## 1.2 Broader context and related work

The work by Perdomo et al. [51], initiated the discourse of performativity in the context of supervised learning by pointing out that the deployment of a predictive model can impact the data distribution we train our models on. Existing scholarship on performative prediction [c.f., 51, 42, 12, 44, 24, 26, 68, 45, 52, 31] has predominantly focused on achieving a particular solution concept with a prediction function that maps $X$ to $Y$ in the presence of unknown performative effects. We are interested in understanding the underlying causal mechanism of the performative distribution shift. Our work is motivated by the seemingly natural approach of lifting the supervised-learning problem and incorporating the prediction as an input feature when building a meta machine learning model for explaining $Y$. By establishing a connection to causal identifiability, our goal is to understand when such a data-driven strategy can help anticipate the down stream effects of predictions

This work focuses on the setting where predictions lead to changes in the relationship between covariates $X$ and label $Y$, while the marginal distribution $P(X)$ over covariates is assumed to be fixed. This setting where performativity only surfaces in the label describes an interesting subclass of problems falling under the umbrella of performative (aka. model-induced or decision-dependent) distribution shifts [51, 37, 12]. Our assumptions are complementary to the strategic classification framework [8, 20] that focuses on a setting where performative effects concern $P(X)$, while $P(Y|X)$ is assumed to remain stable. Consequently, causal questions in strategic classification [e.g., 22, 3, 59] are concerned with identifying stable causal relationships between $X$ and $Y$. Since we assume $P(Y|X)$ can change (i.e. the true underlying 'concept' determining outcomes can change), conceptually different questions emerge in our work. Similar in spirit to strategic classification, the work on algorithmic recourse and counterfactual explanations [32, 28, 65] focuses on the causal link between features and predictions, whereas we focus on the down-stream effects of predictions.

There are interesting parallels between our work and related work on the offline evaluation of online policies [e.g., 35, 63, 36, 58]. In particular, [63] explicitly emphasize the importance of logging propensities of the deployed policy during data collection to be able to mitigate selection bias. In our work the deployed model can induce a concept shift. Thus, we find that additional information about the predictions of the deployed model needs to be recorded to be able to foresee the impact of a new predictive model on the conditional distribution $P(Y|X)$, beyond enabling propensity weighting [55]. A notable work by [66] investigates how predictions at one time step impact predictions in future time steps. Complementary to these existing works we show that randomness in the predictive model is not the only way causal effects of predictions can be identified.

For our theoretical results, we build on classical tools from causal inference [48, 57, 64]. In particular, we distill unique properties of the performative prediction problem to design assumptions for the identifiability of the causal effect of predictions.

## 2 The causal force of prediction

Predictions can be performative and impact the population of individuals they aim to predict. Formulized it in the language of causal inference [48]: the deployment of a predictive model represents an intervention on a causal diagram that describes the underlying data generation process of the population. We will expand on this causal perspective to study an instance of ths performative prediction problem described below.

### 2.1 Prediction as a partial mediator

Consider a machine learning application relying on a predictive model $f$ that maps features $X$ to a predicted label $\hat{Y}$. We assume the predictive model $f$ is performative in that the prediction $\hat{Y} = f(X)$ has a direct causal effect on the outcome variable $Y$ of the individual it concerns. Thereby the prediction impacts how the outcome variable $Y$ is generated from the features $X$. The causal diagram illustrating this setting is visualized in Figure 1.

The features $X \in \mathcal{X} \subseteq \mathbb{R}^d$ are drawn i.i.d. from a fixed underlying continuous distribution over covariates $\mathcal{D}_X$ with support $\mathcal{X}$. The outcome $Y \in \mathcal{Y} \subseteq \mathbb{R}$ is a function of $X$, partially mediated by the prediction $\hat{Y} \in \mathcal{Y}$. The prediction $\hat{Y}$ is determined by the deployed predictive model $f : \mathcal{X} \to \mathcal{Y}$. For a given prediction function $f$, every individual is assumed to be sampled i.i.d. from the data generation process described by the causal graph in Figure 1. We assume the exogenous noise $\xi_Y$ is zero mean, and $\xi_f$ allows the prediction function to be randomized.

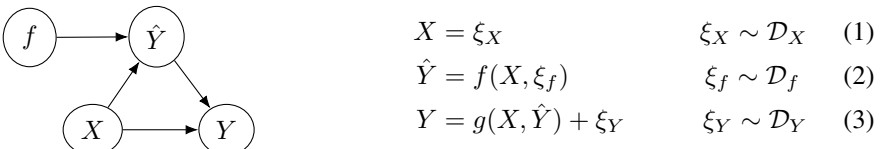

$$X = \xi_X \qquad\qquad \xi_X \sim \mathcal{D}_X \quad (1)$$

$$\hat{Y} = f(X, \xi_f) \qquad\qquad \xi_f \sim \mathcal{D}_f \quad (2)$$

$$Y = g(X, \hat{Y}) + \xi_Y \qquad\qquad \xi_Y \sim \mathcal{D}_Y \quad (3)$$

Figure 1: Performative effects mediated by predictions for a given $f$

Note that our model is not meant to describe performativity in its full generality (which includes other ways $f$ may affect $P(X, Y)$). Rather, it describes an important and practically relevant class of performative feedback problems that are characterized by two properties: 1) performativity surfaces only in the label $Y$, and 2) performative effects are mediated by the prediction, such that $Y \perp\!\!\!\perp f \mid \hat{Y}$, rather than dependent on the specifics of the decision rule.

**Application examples.** Causal effects of predictions on outcomes have been documented in multiple contexts: A bank's prediction about the client (e.g., his or her creditworthiness in applying for a loan) determines the interest rate assigned to them, which in turn changes a client's financial situation [41]. Mathematical models that predict stock prices inform the actions of traders and thus heavily shape financial markets and economic realities [38]. Zillow's housing price predictions directly impact sales prices [39]. Predictions about the severity of an illness play an important role in treatment decisions and hence the very chance of survival of the patient [34]. Another prominent example from psychology is the Pygmalion effect [56]. It refers to the phenomenon that high expectations lead to improved performance, which is widely documented in the context of education [6], sports [61], and organizations [16]. Examples of such performativity abound, and we hope to have convinced the reader that the performative effects in the label are important for algorithmic prediction.

## 2.2 Implications for performativity-agnostic learning

Begin with considering the classical supervised learning task where $\hat{Y}$ is unobserved. The goal is to learn a model $h : \mathcal{X} \to \mathcal{Y}$ for predicting the label $Y$ from the features $X$. To understand the inherent challenge of classical prediction under performativity, we investigate the relationship between $X$ and $Y$ more closely. Specifically, the data generation process (Figure 1) implies that

$$P(Y|X) = \int P(Y|\hat{Y}, X) P(\hat{Y}|X) \mathrm{d}\hat{Y}. \tag{4}$$

This expression makes explicit how the relationship between $X$ and $Y$ that we aim to learn depends on the predictive model governing $P(\hat{Y}|X)$. As a consequence, when the deployed predictive model at test time differs from the model at training time, performative effects surface as concept shift [17]. Such distribution shift problems are known to be intractable without structural knowledge about the shift, implying that we can not expect $h$ to generalize to distributions induced by future model deployments. Let us inspect the resulting extrapolation gap in more detail and put existing positive results on performative prediction into perspective.

**Extrapolation loss.** We illustrate the effect of performativity on predictive performance using a simple instantiation of the structural causal model from Figure 1. Therefore, assume a linear performative effect of strength $\alpha > 0$ and a base function $g_1 : \mathcal{X} \to \mathcal{Y}$

$$g(X, \hat{Y}) := g_1(X) + \alpha\hat{Y}. \tag{5}$$

Now, assume we collect training data under the deployment of a predictive model $f_\theta$ and validate our model under the deployment of $f_\phi$. We adopt the notion of a distribution map from Perdomo et al. [51] and write $\mathcal{D}_{XY}(f)$ for the joint distribution over $(X, Y)$ surfacing from the deployment of a model $f$. We assess the quality of our predictive model $h : \mathcal{X} \to \mathcal{Y}$ over a distribution $\mathcal{D}_{XY}(f)$ induced by $f$ via the loss function $\ell : \mathcal{Y} \times \mathcal{Y} \to \mathbb{R}$ and write $\mathrm{R}_f(h) := \mathrm{E}_{x,y\sim\mathcal{D}_{XY}(f)}\ell(h(x), y)$ for the risk of $h$ on the distribution induced by $f$. We use $h_f^*$ for the risk minimizer $h_f^* := \operatorname{argmin}_{h\in\mathcal{H}} \mathrm{R}_f(h)$, and $\mathcal{H}$ for the hypothesis class we optimize over. Proposition 1 bounds the extrapolation loss and can be viewed as a concrete instantiation of the more general extrapolation bounds for performative prediction discussed in [37] within the feedback model from Figure 1.

**Proposition 1** (Hardness of performativity-agnostic prediction)**.** *Consider the data generation process in Figure 1 with $g$ given in* (5) *and $f_\theta, f_\phi$ being deterministic functions. Take a loss function*

$\ell : \mathcal{Y} \times \mathcal{Y} \to \mathbb{R}$ that is $\gamma$-smooth and $\mu$-strongly convex in its second argument. Let $h^*_{f_\theta}$ be the risk minimizer over the training distribution and assume the problem is realizable, i.e., $h^*_{f_\theta} \in \mathcal{H}$. Then, we can bound the extrapolation loss of $h^*_{f_\theta}$ on the distribution induced by $f_\phi$ as

$$\frac{\gamma}{2}\alpha^2 d^2_{\mathcal{D}_X}(f_\theta, f_\phi) \geq \Delta R_{f_\theta \to f_\phi}(h^*_{f_\theta}) \geq \frac{\mu}{2}\alpha^2 d^2_{\mathcal{D}_X}(f_\theta, f_\phi) \tag{6}$$

where $d^2_{\mathcal{D}_X}(f_\theta, f_\phi) := \mathrm{E}_{x \sim \mathcal{D}_X}(f_\theta(x) - f_\phi(x))^2$ and $\Delta R_{f_\theta \to f_\phi}(h) := \mathrm{R}_{f_\phi}(h) - \mathrm{R}_{f_\theta}(h)$.

The extrapolation loss $\Delta R_{f_\theta \to f_\phi}(h^*_{f_\theta})$ is zero if and only if either the strength of performativity tends to zero ($\alpha \to 0$), or the predictions of the two predictors $f_\theta$ and $f_\phi$ are identical over the support of $\mathcal{D}_X$. If this is not the case, an extrapolation gap is inevitable. This elucidates the fundamental hardness of performative prediction from feature, label pairs $(X, Y)$ when performative effects disrupt the causal relationship between $X$ and $Y$.

The special case where $\alpha = 0$ aligns with the assumption of classical supervised learning, in which there is no performativity. This may hold in practice if the predictive model is solely used for descriptive purposes, or if the agent making the prediction does not enjoy any economic power [21]. The second special case where the extrapolation error is small is when $d^2_{\mathcal{D}_X}(f_\theta, f_\phi) \to 0$. In which case $\mathcal{D}_{XY}(f_\theta)$ and $\mathcal{D}_{XY}(f_\phi)$ are equal in distribution and hence exhibit the same risk minimizer. Such a scenario can happen, for example, if the model $f_\phi$ is obtained by retraining $f_\theta$ on observational data and a fixpoint is reached ($f_\theta = h^*_{f_\theta}$). The convergence of policy optimization strategies to such fixpoints (perfromative stablity) has been studied in prior work [e.g., 51, 42, 12] and enabled optimality results even in the presence of performative concept shifts, relying on the target model $f_\phi$ not being chosen arbitrarily, but based on a pre-specified update strategy.

## 3 Identifying the causal effect of prediction

Having illustrated the hardness of performativity-agnostic learning, we explore under what conditions incorporating the presence of performative predictions into the learning task enables us to anticipate the perfromative effects of $\hat{Y}$ on $Y$. Towards this goal, we assume that the mediator $\hat{Y}$ in Figure 1 is observed—the prediction takes on the role of the treatment in our causal analysis and we can not possibly hope to estimate the treatment effect of a treatment that is unobserved.

### 3.1 Problem setup

Assume we are given access to data points $(x, \hat{y}, y)$ generated i.i.d. from the structural causal model in Figure 1 under the deployment of a prediction function $f_\theta$. From this observational data, we wish to estimate the expected potential outcome of an individual under the deployment of an unseen (but known) predictive model $f_\phi$. We note that given our causal graph, the implication of intervening on the function $f$ can equivalently be explained by an intervention on the prediction $\hat{Y}$. Thus, we are interested in identifying the causal mechanism:

$$\mathcal{M}_Y(x, \hat{y}) := \mathbb{E}[Y | X = x, \mathrm{do}(\hat{Y} = \hat{y})]. \tag{7}$$

Unlike $P(Y|X)$, the mecahnism $\mathcal{M}_Y$ is invariant to the changes in the predictive model governing $P(\hat{Y}|X)$. Thus, being able to identify $\mathcal{M}_Y$ will allow us to make inferences about the potential outcome surfacing from planned model updates beyond explaining patterns in historical data. We can evaluate $\mathcal{M}_Y$ to infer $y$ for any $x$ at $\hat{y} = f_\phi(x)$ for $f_\phi$ being the model of interest.

For simplicity of notation, we will write $\mathcal{D}(f_\theta)$ to denote the joint distribution over $(X, \hat{Y}, Y)$ of the observed data collected under the deployment of the predictive model $f_\theta$. We say $\mathcal{M}_Y$ can be identified, if it can uniquely be expressed as a function of observed data. More formally:

**Definition 1** (identifiability). *Given a predictive model $f$, the causal graph in Figure 1, and a set of assumptions A. We say $\mathcal{M}_Y$ is identifiable from $\mathcal{D}(f)$, if for any function $h$ that complies with assumptions A and $h(x, \hat{y}) = \mathcal{M}_Y(x, \hat{y})$ for pairs $(x, \hat{y}) \in \mathrm{supp}(\mathcal{D}_{XY}(f))$ it must also hold that $h(x, \hat{y}) = \mathcal{M}_Y(x, \hat{y})$ for all pairs $(x, \hat{y}) \in \mathcal{X} \times \mathcal{Y}$.*

Without causal identifiability, there might be models $h' \neq \mathcal{M}_Y$ that explain the training distribution equally well but do not transfer to the distribution induced by the deployment of a new model. Causal identifiability is crucial for enabling extrapolation. It quantifies the limits of what we can infer given access to the training data distribution, ignoring finite sample considerations.

**Identification with supervised learning.** Identifiability of $\mathcal{M}_Y$ from samples of $\mathcal{D}(f_\theta)$ implies that the historical data collected under the deployment of $f_\theta$ contains sufficient information to recover the invariant relationship (7). As a concrete identification strategy, consider the following standard variant of supervised learning that takes in samples $(x, \hat{y}, y)$ and builds a meta-model that predicts $Y$ from $X, \hat{Y}$ by solving the following risk minimization problem

$$h_{\text{SL}} := \underset{h \in \mathcal{H}}{\operatorname{argmin}}\ \mathrm{E}_{(x, \hat{y}, y) \sim \mathcal{D}(f_\theta)}\big[\left(h(x, \hat{y}) - y\right)^2\big]. \tag{8}$$

where $\mathcal{H}$ denotes the hypothesis class. We consider the squared loss for risk minimization because it pairs well with the exogeneous noise $\xi_Y$ in (3) being additive and zero mean. The strategy (8) is an instance of what we term *predicting from predictions*. Lemma 2 provides a sufficient condition for the supervised learning solution $h_{\text{SL}}$ to recover the invariant causal quantity $\mathcal{M}_Y$.

**Lemma 2** (Identification strategy). *Consider the data generation process in Figure 1 and a set of assumptions A. Given a hypothesis class $\mathcal{H}$ such that every $h \in \mathcal{H}$ complies with A and the problem is realizable, i.e., $\mathcal{M}_Y \in \mathcal{H}$. Then, if $\mathcal{M}_Y$ is causally identifiable from $\mathcal{D}(f_\theta)$ given A, the risk minimizer $h_{SL}$ in (8) will coincide with $\mathcal{M}_Y$.*

### 3.2 Challenges for identifiability

The main challenge for identification of $\mathcal{M}_Y$ from data is that in general, the prediction rule $f_\theta$ which produces $\hat{Y}$ is a deterministic function of the covariates $X$. This means that, for any realization of $X$, we only get access to one $\hat{Y} = f_\theta(X)$ in the training distribution, which makes it challenging to disentangle the direct and the indirect effects of $X$ on $Y$. To illustrate this challenge, consider the function $h(x, \hat{y}) := \mathcal{M}_Y(x, f_\theta(x))$ that ignores the input parameter $\hat{y}$ and only relies on $x$ for explaining the outcome. This function explains $y$ equally well and can not be differentiated from $\mathcal{M}_Y$ based on data collected under the deployment of a deterministic prediction rule $f_\theta$. The problem is akin to fitting a linear regression model to two perfectly correlated covariates. More broadly, this ambiguity is due to what is known as a *lack of overlap* (or lack of positivity) in the literature of causal inference [47, 23]. In the covariate shift literature, the lack of overlap surfaces when the covariate distribution violates the common support assumption and the propensity scores are not well-defined (see e.g., Pan and Yang [46]). This problem renders causal identification and thus data-driven learning of performative effects from deterministic predictions fundamentally challenging.

**Proposition 3** (Nonidentifiability from deterministic predictions). *Consider the structural causal model in Figure 1. Assume $Y$ non-trivially depends on $\hat{Y}$, and the set $\mathcal{Y}$ is not a singleton. Then, given a deterministic prediction function $f$, the mechanism $\mathcal{M}_Y$ is not identifiable from $\mathcal{D}(f)$.*

The identifiability issue persists as long as the two variables $X, \hat{Y}$ are deterministically bound and there is no incongruence or hidden structure that can be exploited to disentangle the direct effect of $X$ on $Y$ from the indirect effect mediated by $\hat{Y}$. In the following, we focus on particularities of prediction problems and show how they allow us to identify $\mathcal{M}_Y$.

### 3.3 Identifiability from randomization

We start with the most natural setting that provides identifiability guarantees: randomness in the prediction function $f_\theta$. Using standard arguments about overlap [47] we can identify $\mathcal{M}_Y(x, \hat{y})$ for any pair $x, \hat{y}$ with positive probability in the data distribution $\mathcal{D}(f_\theta)$ from which the training data is sampled. To relate this to our goal of identifying the outcome under the deployment of an unseen model $f_\phi$ we introduce the following definition:

**Definition 2** (output overlap). *Given two predictive models $f_\theta, f_\phi$, the model $f_\phi$ is said to satisfy output overlap with $f_\theta$, if for all $x \in \mathcal{X}$ and any subset $\mathcal{Y}' \subseteq \mathcal{Y}$ with positive measure, it holds that*

$$\frac{\mathrm{P}[f_\phi(x) \in \mathcal{Y}']}{\mathrm{P}[f_\theta(x) \in \mathcal{Y}']} > 0. \tag{9}$$

In particular, output overlap requires the support of the new model's predictions $f_\phi(x)$ to be contained in the support of $f_\theta(x)$ for every potential $x \in \mathcal{X}$. The following proposition takes advantage of the fact that the joint distribution over $(X, Y)$ is fully determined by the deployed model's predictions to relate output overlap to identification:

**Proposition 4.** *Given the causal graph in Figure 1, the mechanism $\mathcal{M}_Y(x, \hat{y})$ is identifiable from $\mathcal{D}(\hat{f}_\theta)$ for any pair $x, \hat{y}$ with $\hat{y} = f_\phi(x)$, as long as $f_\phi$ is a prediction function that satisfies output overlap with $f_\theta$.*

Proposition 4 allows us to pinpoint the models $f_\phi$ to which we can extrapolate to from data collected under $f_\theta$. Furthermore, it makes explicit that for collecting data to learn about performative effects, it is ideal to deploy a predictor $f_\theta$ that is randomized so that the prediction output has full support over $\mathcal{Y}$ for any $x$. Such a model would generate a dataset that guarantees global identification of $\mathcal{M}_Y$ over $\mathcal{X} \times \mathcal{Y}$ and thus robust conclusions about any future deployable model $f_\phi$. One interesting and relevant setting that satisfies this property is the differentially private release of predictions through an additive Laplace (or Gaussian) noise mechanism applied to the output of the prediction function [13].[1]

While standard in the literature, a caveat of identification from randomization is that there are several reasons a decision-maker may choose not to deploy a randomized prediction function in performative environments, including negative externalities and concerns about user welfare [29], but also business interests to preserve consumer value of the prediction-based service offered. In the context of our credit scoring example, random predictions would imply that interest rates are randomly assigned to applicants in order to learn how the rates impact their probability of paying back. We can not presently observe this scenario, given regulatory requirements for lending institutions.

### 3.4 Identifiability through overparameterization

The following two sections consider situations where we can achieve identification, without randomization, from data collected under a deterministic $f_\theta$. Our first result exploits incongruences in functional complexity arising from machine learning models that are overparameterized [e.g. 30]. By overparameterization, we refer to the fact that the representational complexity of the model is larger than the underlying concept it needs to describe.

**Assumption 1** (overparameterization). *We say a function $f$ is overparameterized with respect to $\mathcal{G}$ over $\mathcal{X}$ if there is no function $g' \in \mathcal{G}$ and $c \in \mathbb{R}$ such that $f(x) = c \cdot g'(x)$ for all $x \in \mathcal{X}$.*

A challenge for identification is that for deterministic $f_\theta$ the prediction can be reconstructed from $X$ without relying on $\hat{Y}$, and thus $h(x, \hat{y}) = \mathcal{M}_Y(x, f_\theta(x))$ can not be differentiated from $\mathcal{M}_Y$ based on observational data. However, note that this ambiguity relies on there being an admissable $h$ such that $h(\cdot, \hat{y})$ for a fixed $\hat{y}$ can represent $f_\theta$. If $f_\theta$ is overparameterized with respect to the hypothesis class $\mathcal{H}$, this ambiguity is resolved. Let us make this intuition concrete with an example:

**Example 3.1.** *Assume the structural equation for $y$ in Figure 1 is $g(x, \hat{y}) = \alpha x + \beta \hat{y}$ for some unknown $\alpha, \beta$. Consider prediction functions $f_\theta$ of the following form $f_\theta(x) = \gamma x^2 + \xi x$ for some $\gamma, \xi \geq 0$. Consider $\mathcal{H}$ be the class of linear functions. Then, any admissable estimate $h \in \mathcal{H}$ takes the form $h(x, \hat{y}) = \alpha' x + \beta' \hat{y}$. For $h$ to be consistent with observations we need $\alpha' + \beta' \xi = \alpha + \beta \xi$ and $\beta' \gamma = \beta \gamma$. This system of equations has a unique solution as long as $\gamma > 0$ which corresponds to the case where $f_\theta$ is overparameterized with respect to $\mathcal{H}$. In contrast, for $\gamma = 0$ the function $h(x, \hat{y}) = (\alpha + \beta \xi)x$ would explain the training data equally well.*

The following result generalizes this argument to separable functions.

**Proposition 5.** *Consider the structural causal model in Figure 1 where $f_\theta$ is a deterministic function. Assume that $g$ can be decomposed as $g(X, \hat{Y}) = g_1(X) + \alpha \hat{Y}$ for some $\alpha > 0$ and $g_1 \in \mathcal{G}$, where the function class $\mathcal{G}$ is closed under addition (i.e. $g_1, g_2 \in \mathcal{G} \Rightarrow a_1 \cdot g_1 + a_2 \cdot g_2 \in \mathcal{G} \quad \forall a_1, a_2 \in \mathbb{R}$). Let $\mathcal{H}$ contain functions that are separable in $X$ and $\hat{Y}$, linear in $\hat{Y}$, and $\forall h \in \mathcal{H}$ it holds that $h(\cdot, \hat{y}) \in \mathcal{G}$ for a fixed $\hat{y}$. Then, if $f_\theta$ is overparameterized with respect to $\mathcal{G}$ over the support of $\mathcal{D}_X$, $\mathcal{M}_Y$ is identifiable from $\mathcal{D}(f_\theta)$.*

### 3.5 Identifiability from classification

A second ubiquitous source of incongruence that we can exploit for identification is the *discrete* nature of predictions in the context of classification. The resulting discontinuity in the relationship between $X$ and $\hat{Y}$ enables us to disentangle $\mathcal{M}_Y$ from the direct effect of $X$ on $Y$. This identification strategy is akin to the popular regression discontinuity design [33] and relies on the assumption that all other variables in $X$ are continuously related to $Y$ around the discontinuities in $\hat{Y}$.

---

[1]In Appendix B we discuss two additional natural sources of randomness (randomized decisions and noisy measurements of covariates) that can potentially help identification with appropriate side-information.

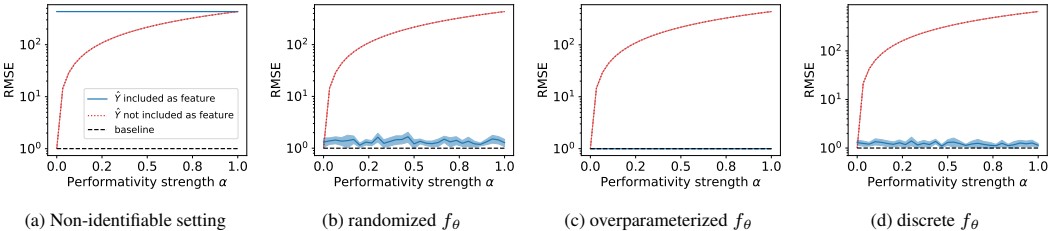

(a) Non-identifiable setting     (b) randomized $f_\theta$     (c) overparameterized $f_\theta$     (d) discrete $f_\theta$

Figure 2: Extrapolation error of supervised learning with and without access to $\hat{Y}$. (a) In the non-identifiable setting, adding $\hat{Y}$ as a feature harms generalization performance. (b)-(d) Randomization, overparameterization, and discrete predictions are each sufficient for avoiding this failure mode.

**Proposition 6.** *Consider the structural causal model in Figure 1 where $f_\theta$ is a deterministic function. Assume that the structural equation for $Y$ is separable $g(X, \hat{Y}) = g_1(X) + g_2(\hat{Y}), \forall X, \hat{Y}$ for some differentiable functions $g_1$ and $g_2$. Further, suppose $X$ is a continuous random variable and $\hat{Y}$ is a discrete random variable that takes on at least two distinct values with non-zero probability. Then, $\mathcal{M}_Y$ is identifiable from $\mathcal{D}(f_\theta)$.*

Similar to Proposition 5, the separability assumption together with incongruence provides a way to disentangle the direct effect from the indirect effect of $X$ on $Y$. Separability is necessary in order to achieve global identification guarantees without randomness, the identification of entangled components without overlap is fundamentally hard. Thus, under violations of the separability assumptions, we can only expect the separable components of $g$ to be correctly identified. Similarly, a regression discontinuity design only enables the identification of the causal effect locally around the discontinuity. Extrapolation away from the decision boundary to models $f_\phi$ that are substantially different from $f_\theta$ increasingly relies on separability to hold true.

## 4 Empirical evaluation

We investigate empirically how well the supervised learning solution $h_{\text{SL}}$ in (8) is able to identify the causal mechanism $\mathcal{M}_Y$ from observational data in practical settings with finite data.

**Methodology.** We generated semi-synthetic data for our experiments, using a Census income prediction dataset from `folktables.org` [11]. Using this dataset as a starting point, we simulate a training dataset and test dataset with distribution shift as follows: First, we choose two different predictors $f_\theta$ and $f_\phi$ to predict a target variable of interest (e.g. income) from covariates $X$ (e.g. age, occupation, education, etc.). If not specified otherwise, $f_\theta$ is fit to the original dataset to minimize squared error, while $f_\phi$ is trained on randomly shuffled labels. Next, we posit a function $g$ for simulating the performative effects. Then, we generate a *training* dataset of $(X, \hat{Y}, Y)$ tuples from the causal model in Figure 1, using the covariates $X$ from the original data, $g$, and $f_\theta$ to generate $\hat{Y}$ and $Y$. Similarly, we generate a *test* dataset of $(X, \hat{Y}, Y)$ tuples, using $X, g, f_\phi$. We assess how well supervised methods learn transferable functional relationships by fitting a model $h_{\text{SL}}$ to the training dataset and then evaluating the root mean squared error (RMSE) for regression and the accuracy for classification on the test dataset. In our figures, we visualize the standard error from 10 replicates with different random seeds and we compare it to an in-distribution baseline trained and evaluated on samples of $\mathcal{D}(f_\phi)$. If not specified otherwise we use $N = 200,000$ samples.

### 4.1 Necessity of identification guarantees for supervised learning

We start by illustrating why our identification guarantees are crucial for supervised learning under performativity. Therefore, we instantiate the structural equation $g$ in Figure 1 as

$$g(X, \hat{Y}) = g_1(X) + \alpha\hat{Y} \tag{10}$$

with $g_1(X) = \beta^\top X$ and $\xi_Y \sim \mathcal{N}(0, 1)$. The coefficients $\beta$ are determined by linear regression on the original dataset. The hyperparameter $\alpha$ quantifies the performativity strength that we vary in our experiments. The predictions $\hat{Y}$ are generated from a linear model $f_\theta$ that we modify to illustrate the resulting impact on identifiability. We optimize $h_{\text{SL}}$ in (8) over $\mathcal{H}$ being the class of linear functions.

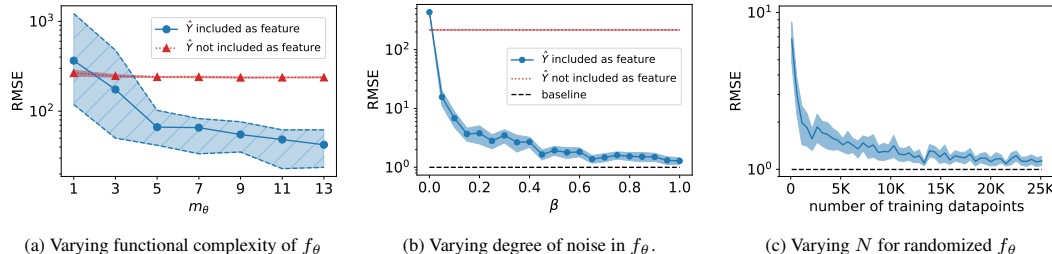

(a) Varying functional complexity of $f_\theta$     (b) Varying degree of noise in $f_\theta$.     (c) Varying $N$ for randomized $f_\theta$

Figure 3: Ablation study of extrapolation performance. (a) We vary $m_\theta$. Adding $\hat{Y}$ as a feature helps as soon as $f_\theta$ is overparameterized with respect to $g_1$ ($m_\theta > 3$). (b) We vary the noise in the predictions of $f_\theta$. A small amount of noise is sufficient for identifiability. (c) We vary number of datapoints for training $h_{\mathrm{SL}}$.

We start by illustrating a failure mode of supervised learning in a non-identifiability setting (Proposition 3). Therefore, we let $f_\theta$ be a deterministic linear model fit to the base dataset ($f_\theta(X) \approx \beta^\top X$). This results in $\mathcal{M}_Y$ not being identifiable from $\mathcal{D}(f_\theta)$. In Figure 2(a) we can see that supervised learning indeed struggles to identify a transferable functional relationship from the training data. The meta model returns $h_{\mathrm{SL}}(X, \hat{Y}) = (1 + \alpha)\hat{Y}$, instead of identifying $g$, which leads to a high extrapolation error independent of the strength of performativity. While we only show the error for one $f_\phi$ in Figure 2(a), the error grows with the distance $d^2_{\mathcal{D}_x}(f_\theta, f_\phi)$. In contrast, when the feature $\hat{Y}$ is not included, the supervised learning strategy returns $h_{\mathrm{SL}}(X) = (1 + \alpha)\beta^\top X$. The extrapolation loss of this performativity-agnostic model scales with the strength of performativity (Proposition 1) and is thus strictly smaller than the error of the model that predicts from predictions.

Next, we move to the regime of our identification results (Proposition 4-6). Therefore, we modify the way the predictions in the training data are generated. In Figure 2(b) we use additive Gaussian noise to determine the predictions as $\hat{Y} = f_\theta(X) + \eta$ with $\eta \in \mathcal{N}(0, \sigma^2)$. In Figure 2(c) we augment the input to $f_\theta$ with second-degree polynomial features to achieve overparameterization. In Figure 2(d) we round the predictions of $f_\theta$ to obtain discrete values. In all three cases, including $\hat{Y}$ as a feature is beneficial and allows the model to match in-distribution accuracy baselines, closing the extrapolation gap that is inevitable for performativity-agnostic prediction.

### 4.2    Strength of incongruence and finite samples

We next conduct an ablation study and investigate how the degree of overparameterization and the noise level for randomized $f_\theta$ impacts the extrapolation performance of supervised learning. Therefore, we consider the setup in (10) with a general function $g_1$. We fix the level of performativity at $\alpha = 0.5$ for this experiment. We optimize $h_{\mathrm{SL}}$ in (8) over $\mathcal{H}$ (which we vary).

In Figure 3(a) we investigate the effect of overparameterization of $f_\theta$ on the extrapolation error of $h_{\mathrm{SL}}$. We choose fully connected neural networks with a single hidden layer to represent the functions $g_1$, $f_\theta$ and $h_{\mathrm{SL}}$. For $g_1$ and $\mathcal{H}$ we take a neural network with $m = 3$ units in the hidden layer. The model $g_1$ is fit it to the original dataset. We vary the number of units in the hidden layer of $f_\theta$, denoted $m_\theta$. As expected, the extrapolation error decreases with the complexity of $f_\theta$. As soon as $m_\theta > m_\phi$ there is a significant benefit to including predictions as features. In this regime, $\mathcal{M}_Y$ becomes identifiable as Proposition 5 suggests. In turn, without access to $\hat{Y}$ the model suffers an inevitable extrapolation gap due to a concept shift that is independent of the properties of $f_\theta$. In Figure 2(b) we investigate the effect of the *magnitude of additive noise* added to the predictions. Here $\mathcal{H}$ and $g_1$ are linear functions. We have $\hat{Y} = f_\theta(X) + \beta\eta$ with $\eta \in \mathcal{N}(0, 1)$ and we vary the noise level $\beta$. We see that even small amounts of noise are sufficient for identification and adding $\hat{Y}$ as a feature to our meta-machine lenaring model is effective as soon as the noise in $f_\theta$ is non-zero. In Figure 2(c) we fix the noise level at $\alpha = 0.5$ and vary the number of samples $N$. We find that only moderate dataset sizes are necessary for predicting from predictions to approximate $\mathcal{M}_Y$ in our identifiable settings.

## 5   Discussion

This paper focused on identifying the causal effect of predictions on outcomes from observational data. We point out several natural situations where this causal question can be answered, but we

also highlight situations where observational data is not sufficiently informative to reason about performative effects. By establishing a connection between causal identifiability and the feasibility of anticipating performative effects using data-driven techniques, this paper contributes to a better understanding of the suitability of supervised learning techniques for explaining social effects arising from the deployment of predictive models in economically and socially relevant applications.

We hope the positive results in this work serve as a *message for data-collection*: only if predictions are observed, they can be incorporated to anticipate the performative effects of future model deployments. Thus, access to this information is crucial for an analyst hoping to understand the effects of deployed predictive models, an engineer hoping to foresee consequences of model updates, or a researcher studying performative phenomena. To date, such data is scarcely available in benchmark datasets, hindering the progress towards a better understanding of performative effects, essential for the reliable deployment of algorithmic systems in the social world.

At the same time we have shown that the deterministic nature of prediction poses unique challenges for causal identifiability even if $\hat{Y}$ is observed. Thus, the success of observational designs (as shown in our empirical investigations) is closely tied to the corresponding identifiability conditions being satisfied. Our results must not be understood as a green-light to justify the use of supervised learning techniques to address performativity in full generality beyond the scope of our theoretical results.

**Limitations and Extensions.**    The central assumption of our work is the causal model in Figure 1. While carving out a rich and interesting class of performative prediction problems that allows us to articulate the challenges of covariates and predictions being coupled, it can not account for all mechanisms of performativity. This in turn gives rise to interesting questions for follow-up studies.

A first neglected aspect is *performativity through social influence*. Our causal model, relies on the stable unit treatment value assumption (SUTVA) [23]. There is no possibility for the prediction of one individual to impact the outcome of his or her peers. Such an individualistic perspective is not unique to our paper but prevalent in existing causal analyses and model-based approaches to performative prediction and strategic classification [e.g., 20, 25, 43, 3, 18, 22]. Spillover effects [cf. 60, 64, 1, 40] are yet unexplored in the context of performative prediction. Nevertheless, they have important implications for how causal effects should be estimated and interpreted. In the context of our work they imply that an intervention on $f$ can no longer be explain solely by changing an individual's prediction. As a result, approaches for microfounding performative effect based on models learned from simple, unilateral interventions on an individual's prediction result in different causal estimates than supervised learning based methods for identification as studied in this work. A preliminary study included in Appendix C shows that data-driven techniques can pick up on interference patterns in the data and benefit from structural properties such as network homophily [19], whereas individualistic modeling misses out on the indirect component arising from neighbors influencing each other.

A second aspect is *performativity in non-causal prediction.* Our model posits that prediction is solely based on features $X$ that are causal for the outcome $Y$. This is a desirable situation in many practical applications because causal predictions disincentivize gaming of strategic individuals manipulating their features [43, 3] and offer explanations for the outcome that persist across environments [54, 7]. Nevertheless, non-causal variables are often included as input features in practical machine learning prediction tasks. Establishing a better understanding for the implications of the resulting causal dependencies due to performativity could be an important direction for future work.

Finally, performative effect can also lead to *covariate shift* and impact the joint distribution $P(X, Y) = P(Y|X)P(X)$ over covariates and labels. We assumed that performative effects only surface in $P(Y|X)$. For our theoretical results, this implied that overlap in the $X$ variable across environments is trivially satisfied, which enabled us to pinpoint the challenges of learning performative effects due to the coupling between $X$ and $\hat{Y}$. For establishing identification in the presence of a causal arrow $f_\theta \to X$ additional steps are required to ensure identifiability.

**Acknowledgement**

The authors would like to thank Moritz Hardt and Lydia Liu for many helpful discussions throughout the development of this project, Tijana Zrnic, Krikamol Muandet, Jacob Steinhardt, Meena Jagadeesan and Juan Perdomo for feedback on the manuscript, and Gary Cheng for helpful discussions on differential privacy. We are also grateful for a constructive discourse and valuable feedback provided by the reviewers that greatly helped improve the manuscript.

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
