# OpenReview forum: "Anticipating Performativity by Predicting from Predictions"
_NeurIPS.cc/2022/Conference — NeurIPS 2022 Accept_

### Official Review · Reviewer_JT8c · 2022-07-08

**Rating:** 7
**Confidence:** 4
**Soundness:** 2 fair
**Presentation:** 4 excellent
**Contribution:** 3 good

**Summary:**

In the traditional supervised setting, we seek to predict a response $Y$ based on features $X$, say as $\hat{Y} = f_{\theta}(X)$. The question this paper asks is the following: What happens when the prediction $\hat{Y}$ itself influences $Y$? Can we then still rigorously reason about our predictive model? To answer the above question, the paper makes the following contributions:

1) It introduces a causal framework to answer the above question. Instead of the traditional regression estimand $\mathbb E[Y \mid X=x]$, the estimand of interest in this setting is the following:
$$ h(x, \hat{y}) = \mathbb E[ Y \mid X=x, do(\hat{Y}=\hat{y})] $$

2) It establishes three practical situations under which the above estimand is identified from observational data and may furthermore be estimated by supervised learning based on weighted empirical risk minimization. These three situations are the following: when noisy predictions are released (e.g., in differential privacy), when the features $X$ are noisy measurements of latent features $U$, and under separability (along with overparameterization or discrete classification).


3) A semi-synthetic study is conducted based on census data and Kaggle credit scoring data.


**Questions:**

### Extrapolation error

Please clearly define what extrapolation error is (I do not think it is ever defined). It may be helpful to state the theoretical result (Proposition 1) with the middle term in the inequality replaced by $R_{f_{\phi}}(\psi(f_{\theta})) - R_{f_{\phi}}(\psi(f_{\phi}))$. Even though the 2nd term is $0$, it may be more instructive to express it as above. In Proposition 1, it seems that smoothness and strong convexity refer to the second argument of the loss function. If so, this should be clarified/explicitly stated.

### Overlap

* Front door adjustment: I did not understand the reference to front door adjustment in both the main text and the proof of Proposition 3. For example, in the proof, it is written that the front door adjustment formula is applied to $\hat{Y} \to T_{\hat{Y}} \to Y$. But note that here $X$ is actually observed, and is conditioned upon, so front door adjustment does not seem necessary? Instead it seems that the argument boils down to the fact that in this setting:
$$ \mathbb E[ Y \mid X=x, do(\hat{Y}=\hat{y})]  = \mathbb E[ Y \mid X=x, \hat{Y}=\hat{y}],$$
and the RHS is now estimable/identifiable (while is is not under the conditions of Proposition 2).


* Page 6, Line 226: $P(T_{\hat{Y}} = t \mid X)>0$: This is not correct, this should be replaced by a statement about the conditional density of $T_{\hat{Y}}$, or alternatively, stated as in Assumption 1.

### Noisy measurement of covariates

* "information about the noise distribution of the measurement error $P(U \mid X=x)$": Is the measurement error distribution not specified instead by $p(x \mid U=u)$?

* I do not understand the statement of Proposition 4. What are these weights exactly? They are stated as:
$w(x, \hat{y}) = P(U | X=x) / P(U \mid \hat{Y} = \hat{y}),$
but this does not make sense to me, since $U$ is a latent random variable (and so the above could not be the density at a fixed $u$). Also why does the beginning of the statement state that knowledge of $P(U \mid \hat{Y} = \hat{y}, Y=y)$ is required even though only $P(U \mid \hat{Y} = \hat{y})$ is included in the weight definition?

* Proposition 4: In the proof of Proposition 4, the equality in line (19) is incorrect.

### Separability:

* Proposition 5: It would be very helpful if the specification of $\mathcal{H}$ in this statement could be more explicit.

* Example 3.1: "inferring $\beta = c_1$ and $\alpha = c_1 - c_2$": I think for $\alpha$ one would infer $\alpha=c_1$ instead of $\alpha = c_1 - c_2$.

* Proposition 6: Here the main text claims that results for the regression discontinuity design are used, but then the actual proof cites a result from Wang and Blei (2019) that is unrelated to the regression discontinuity design. Also in the proof, what is $x'$? Why does supervised learning recover the true effect?

### Minor:

* *The supervised learning approach*: This should be in bold instead of italics (to match the bold of other paragraphs in the manuscript).

* In equation (8), why are the weights allowed to depend also on $y$? Would it suffice if they only depend on $(x, \hat{y})$?

* There is a typo in the statement of Assumption 2.

**Limitations:**

Limitations of the framework are not discussed. It would be great if a paragraph could be added with possible directions for future work!

**Strengths And Weaknesses:**

# Strengths

* Motivation: The setting and problem tackled in this paper are exceptionally well-motivated. It is clear that rigorous reasoning about the causal effects of predictions is of great practical and theoretical importance.

* Clarity: The prose of the paper (but not the formal results, see below) is a joy to read and very clear.


# Weaknesses

* Previous work: I believe that the paper misses important previous references. Rigorously thinking about causal effects of predictions and seeking to estimate them is not a new idea, even within the NeurIPS community. For example, the following paper (also see references therein) addresses such a problem, and the underlying method is based on adding noise to the predictors (i.e., very similar to one of the identification strategies in the present paper):

> Wager, S., Chamandy, N., Muralidharan, O., & Najmi, A. (2014). Feedback detection for live predictors. Advances in Neural Information Processing Systems, 27.

* Formal results: Several of the formal results of the paper are not well-stated, have errors (e.g., in the proofs), and make incorrect references to previous work. See points below for a more detailed listing of such issues.

---

> ### Author Response · Authors · 2022-08-02
> **Author response**
>
> **Related work.** Thank you for the relevant pointer. The work by Wager et al paper is interesting because it is also motivated by the downstream effects of predictions, however the feedback effects studied are different from the focus of our work. More specifically, they assume repeated prediction, access to data from multiple consecutive time steps, and focus on detecting whether predictions in one time step impact predictions in the next time step. They show that for detecting such feedback loops, randomness in predictions is sufficient. But apart from the fact that independent variation in predictions helps reason about down-stream consequences of predictions, the insights of the paper are different from ours. In particular, detecting such feedback loops is not sufficient for concluding specifics of how predictions impact individuals, which is what we are interested in. We will elaborate on this.
>
> **Theoretical statements.** There are some inaccuracies in the statement of the theoretical results in the submitted version. We are aware of them and we thank the reviewer for carefully going through the claims and pointing them out. We have fixed them in the meantime. Let us comment on the specific questions:
>
> - *Extrapolation error.* Yes, the conditions are both assumed to hold in the second argument of the loss function. We will clarify this and also add a formal definition of the extrapolation error.
> - *Front-door-adjustment.* We agree that technically there is no confounding in our setting. We used the term font-door to highlight the conceptual approach of causal identification by tracing the path of the causal effect rather than by adjusting. We will clarify this.
> - *Overlap condition.* Thanks for spotting this.  $P(T_{\hat Y}=t∣X)>0$ should be replaced with a condition over measurable sets to make sure it is well defined for continuous t.
> - *Example 3.1* We restated the assumption in the example as follows:  Any consistent parameter estimate $\alpha',\beta'$ needs to satisfy the following two equations: $ \alpha'+\beta'\xi = \beta\xi$ and $\beta'\gamma=\beta\gamma$ matching the multipliers of the base polynomials. It is not hard to see that this system of equations has a unique solution as long as $\gamma>0$ which corresponds to the case where $f_\theta$ is overparameterized with respect to the class of linear functions.
> - *Proposition 4.* Knowledge of $P(U∣\hat Y=\hat y,Y=y)$ is required because samples need to be drawn from this distribution in order for the ERM statement to hold with the specified weights. This can directly be seen from Equation (23) in the appendix. So there is some additional adjustment that needs to be done to map the result to the ERM problem, it is not as straightforward as we presented it in the paper.  We have decided to remove this result from the paper because the identification strategy is very different from the other results, it does not neatly fit with the ERM problem. In our eyes it does not add substantially to the story and we prefer to leave the development of more complex identification strategies for future work. Instead we have added a refined randomness result to make explicit the set of models we can extrapolate to even if global identifiability can not be established.
> - *Regression discontinuity*. We make an analogy to RDD because we believe many people are familiar with this quasi-experiment strategy and the reasoning behind it. However, Wang and Blei (2019) was the result that we relied on in the proof, thus the reference. Conceptually the results are very similar in spirit. Wang and Blei (2019) leverages a similar inductive bias as the regression discontinuity design to achieve causal identification; they both rely on the incongruence between the discrete treatment and continuous confounder for identification. The $x’$ in the proof refers to any $x’\neq x$ such that $(\hat y,x’)$ is observed, we will clarify this.
>
> We hope this discussion clarifies the concerns with the theoretical statements. Please let us know if these answers are not satisfactory.

---

> > ### Comment · Reviewer_JT8c · 2022-08-08
> > **Response**
> >
> > Thank you for your response. I have increased my score based on the changes promised for a camera-ready version of this paper.
> >
> > Regarding $P(U∣\hat Y=\hat y,Y=y)$, I still do not understand what this refers to (since $U$ is unobserved) but perhaps this does not matter if this result is removed from the manuscript.

---

> > > ### Author Response · Authors · 2022-08-09
> > > **follow-up**
> > >
> > > We appreciate your positive assessment and the helpful comments. We will for sure implement all these changes in the camera-ready version to the best of our abilities.
> > >
> > > Regarding your last question:
> > > the fact that $U$ is unobserved means that, for each individual, we do not observe their precise value of $U$, but only observe $X, \hat Y, Y$. However, we may still have knowledge about the distribution of the measurement noise $P(U|X)$ which enables the application of our result. For example $P(U|X) = N(X, \sigma^2)$ where $\sigma$ indicates the scale of the noise. (Knowing the size of noise does not reveal the precise value of U either.) Depending on the application such external knowledge about the noise may be available for example from test–retest data, prior modeling, or a physical model for the measurement device.
> > > Given such external knowledge about noise, we can calculate the distribution $P(U|\hat Y, Y) = \int P(U|X) P(X|\hat Y,Y) dX$ even when $U$ is unobserved; $P(U|X)$ comes from the external knowledge about measurement noise; and $P(X|\hat Y,Y)$ is a distribution about observed variables and may be estimated from data.
> > >
> > > We hope this clarifies your question even if we agree it is less relevant for the updated manuscript.

---

### Official Review · Reviewer_Lu7e · 2022-07-09

**Rating:** 6
**Confidence:** 4
**Soundness:** 4 excellent
**Presentation:** 4 excellent
**Contribution:** 3 good

**Summary:**

The paper studies the problem of estimating counterfactual outcomes under a different predictive model, thus drawing a link between causal inference and the recent literature on performative prediction. The paper starts with hardness of performativity-agonostic learning and then describes identification results that enable the supervised learning approach to work. The authors also conduct experiments showing that the identification strategies work well in simulations and mild violations of assumptions are not of concern.

**Questions:**

1. In the overparametrization identification result, the author mentions neural network as one of the examples. I am curious about if there is any empirical results illustrating this identification result under this complex model as a lot of the real world models are indeed getting more and more complex.
2. In proposition 4, is $P(U|Y, \hat{Y})$ necessary to know or $P(U|\hat{Y})$ is enough to enable identification?
3. Could you say more about applicability of knowing $P(U|Y, \hat{Y})$ in some real world examples, like which examples you mentioned seem more reasonable assuming this?

Some possible typos: line 293 in main text, $f_\theta$ we will, removing we? line 307, should there be $Y_j$ after the expectation?

**Limitations:**

The author is very clear about the limitation of the work.

**Strengths And Weaknesses:**

Strengths: The writing is clear and the problem is well motivated and stated. The identification results are clearly stated with enough explanations and examples. Overall the paper is very clear in exposition. The problem studied is also having potential significance in real world examples as non-stationarity arises, which I particuarly like. The experiements are sound with code available for reproducibility.

Weakness: The scope of the work is somewhat limited since the effect of the predictive model is only assumed to affect $
\hat{Y}$ (the authors acknowledge this in section 2.1). When restricting to this particular causal graph, the first two identification results are essentially direct application of existing results, thus limiting the originality of the work. Also, it would be nice to see some real examples in the experiment section.

---

> ### Author Response · Authors · 2022-08-02
> **Author response**
>
> **Novelty of the theoretical results.** We agree that, technically, the first two results are direct applications of existing results from causal inference to the performative prediction problem. However we believe there is value in singling out the causal question, transferring insights between these two fields and interpreting the standard assumptions in the context of prediction. This illuminates interesting connections to differential privacy and individual fairness. Furthermore, to better integrate the result Section 3.2 with the overall story we added a general identification results, stating the positivity assumption as an assumption on the predictive models, rather than the induced distribution directly. This builds on the unique property of the performative prediction problem where distributions are characterized by the predictions of the deployed model, and further puts it apart from standard causal identifiability results.
>
> **Overparameterization in neural networks.** Yes, it is possible to replace the overparameterized $f_\theta$ in our experiments in Figure 4(c) with a neural network instead of just a degree-2 polynomial. That’s a good idea. We can also simultaneously increase the complexity of $g_1$ and $g_2$ while complying with the assumptions of Proposition 6. We will add this experiment to the appendix.
>
> **Proposition 4.** We need to know $P(U|\hat Y)$ for computing the weights but we also need to know  $P(U|Y,\hat Y)$ because this is the distribution we need to sample from in order to achieve identification. This can be seen from equation (23) in the appendix. Because of this change in distribution, the correspondence with the ERM objective is a bit more involved than stated in the paper. We have decided to remove this result from the main body and focus on results that enable identification via the standard ERM framework. We leave the discussion of more complex identification strategies for future work. To us the most interesting results are Proposition 5&6 that show how two natural properties of prediction functions can help identification despite lack of overlap and we do not want technical details to distract from this clear story. Together with the refined randomness result we believe they can convey a complete picture.
>
> Thanks for spotting the typos.

---

> > ### Comment · Reviewer_Lu7e · 2022-08-07
> > **Reviewer Response**
> >
> > Thanks for the detailed response. I still think the identification results are not novel enough and very specific to the causal structure. And it would be better to move proposition 4 to appendix rather than deleting it given that you only have two identification results now. But I think the extra experiments and the clarifications you made here as well as in other responses compensate this weakness so I tend to remain the evaluation I had.

---

> > > ### Author Response · Authors · 2022-08-08
> > > **Follow-up**
> > >
> > > Thank you for your feedback! We are happy to have Proposition 4 in the appendix where we have more space to elaborate on the details, we just felt it is hard to do justice to this result in the main body. We believe it serves as an interesting example of a possible identification strategy with side-information that goes beyond classical ERM.
> > >
> > > We would like to emphasize that we will still have *four* identification results in the paper: identification through 1)* randomization in predictions and 2) randomization in down-stream decisions, and the two identification results without randomness through 3) incongruence in functional complexity (overparameterization) and 4) incongruence in modality (discrete prediction). We don't think the first two should be dismissed because they are 'technically' less evolved; they connect properties of predictions to overlap from where identification follows. The former connection has not previously been made in the context of performative prediction. We think understanding the limitations of learning performative effects from data collected under the deployment of a *deterministic predictions* and illustrating the need for sufficient randomness (as well as discussing natural settings where it arises) is very relevant in practice.
> > >
> > > *The first result is the refined randomness result we decided to add in response to the reviewer comments. It is outlined in the general response, bullet point 4.

---

> > > > ### Comment · Reviewer_Lu7e · 2022-08-09
> > > > **Follow-up**
> > > >
> > > > Thanks for the clarification!

---

### Official Review · Reviewer_rWDi · 2022-07-10

**Rating:** 4
**Confidence:** 4
**Soundness:** 3 good
**Presentation:** 2 fair
**Contribution:** 1 poor

**Summary:**

This work investigates performative prediction through the lens of causal inference, aiming to identify causal structures which allow for counterfactual estimation of performative effects.

Investigation in this paper assumes a specific causal structure, in which the predicted value $\hat{Y}$ acts as a mediator between the chosen prediction model $f_\theta$ and the outcome $Y$, and $f_\theta$ does not directly affect other variables (Figure 1). For the theoretical analysis, an infinite-sample setting is assumed, where it is assumed that the sample size is infinite.
Three sets of theoretical results are presented:
* The first set result assumes a specific performative structure, and quantifies the regret from naively training a model under performative behavior (Proposition 1,2). The result aims to illustrate the negative effects of neglecting a performative causal structure.
* For the second set of theoretical results, authors identify “overlap/positivity” as the main limiting factor in performative extrapolation. Authors point out three avenues through which this concern can be alleviated - randomization of model outputs (Proposition 3), noisy measurement of covariates (Proposition 4), and incongruence between the effect of $X$ on $Y$ and the effect of $X$ on $\hat{Y}$ in separable performative structures (“over-parameterization” in Proposition 5, discrete classification in Proposition 6).
* For the third set of results, authors initiate investigation of spillover effects between users, identifying a spillover structure $G$ through which identifiability is possible.

In the experimental evaluation, authors present results which validate their claims, showing empirically that results extend beyond the theoretical guarantees.


**Questions:**

* When do we expect the performative causal structure assumption (Figure 1) to hold? When do we expect it not to hold? What are the implications of making a wrong assumption about the causal structure?
* What is the relation between the different propositions in the theoretical analysis? How do they combine into an understanding of a “bigger picture”? Are they "complete" in the sense that they cover all possible influence modes?
* The discussion claims that “one of the most important lessons from this work is that there is high value to logging the state of the deployed predictor when collecting data for the purpose of supervised learning”. I agree with this claim very much, and am wondering if there is a way to support or quantify it.
* What is the connection to existing work on learning under distribution shift? Literature that comes to mind is:
  * “Counterfactual Risk Minimization: Learning from Logged Bandit Feedback” by Swaminathan and Joachims 2015,
  * “Discriminative Learning Under Covariate Shift”, Bickel et al. 2009
  * “Actionable Recourse in Linear Classification”, Uston et al. 2018
Literature on online learning and multi-armed bandits (e.g can we think of an “$\varepsilon$-greedy” strategy as an inducer of identifiability?)


**Limitations:**

* The paper makes a very strong causal assumption, but the current version does not properly contextualize it within the existing literature. I feel that the paper would benefit from discussing its limitations and implications of the causal structure assumption in more depth.
* Extension of theoretical analysis to the finite-sample case, and the relation to existing machine learning literature on similar topics.

**Strengths And Weaknesses:**

Strengths
* Presentation and mathematical notations are very clear. Causal assumptions are made explicit.
* Identifying “failure modes” and “success modes” of performative prediction is an approach which can establish strong theoretical foundations for this area of research, and help it further establish applicability in practice.
* The investigation points out avenues of research which may be interesting for further inquiry, such as the relation between model over-parameterization and performative prediction, or the significance of spillover effects.

Weaknesses
* The paper claims to address the general problem of "predicting from predictions", but in practice assumes a very specific causal structure and does not sufficiently establish its applicability. In particular, the paper assumes a causal model in which $f_\theta$ does not affect $X$. However, a direct causal path between $f_\theta$ and $X$ exists in many practical cases, for example in the “Actionable Recourse” setting (Uston et al. 2018). In section 2.1, the authors mention many applications in which a performative structure may be present, but it’s not clear whether they can indeed be approximated using the causal structure presented in Figure 1.
* The paper presents a collection of interesting insights, but it’s not made clear how they combine into an understanding of the “bigger picture”. Moreover, even though similar problems were investigated in previous literature (e.g “Counterfactual Risk Minimization: Learning from Logged Bandit Feedback” by Swaminathan and Joachims 2015, recent work on over-parameterization and OoD generalization in deep neural nets), these existing results are not discussed in the paper.
* Theoretical results assume that sample size is infinite, and that predictors are minimizers of the expected risk (i.e $h^*=\arg\min_{h\in\mathcal{H}} \mathbb{E}[w(h(x)-y)^2]$) - A quantity which cannot be directly optimized practice. Not clear how the formalism extends to the finite-sample setting. What would change when the dataset size is finite?

A few minor typos: Line 263 (bet -> be), line 307 (inequality expression seems incomplete - missing a random variable next to the expectation operator?), line 374 (lesson -> lessons).

---

> ### Author Response · Authors · 2022-08-02
> **Author response**
>
> **Causal model.** This work aims to draw attention to the direct causal link between predictions and outcomes. This link reflects that the deployment of a model can lead to changes in P(Y|X). We chose to single out this effect because it is often assumed away by making a covariate shift assumption in out-of-distribution generalization. Certainly, the causal link $f\rightarrow\hat Y\rightarrow Y$ is not the only way performative effects in the joint distribution $P(X,Y)$ surface in practice. We aimed to make this explicit by contrasting with the strategic classification setup in the related work, but we can expand on this (e.g., actional recourse is also an interesting example that is concerned with the link between $X$ and $\hat Y$).
>
> While causal identifiability claims are no longer valid out-of-the-box if additional causal arrows are added, the conceptual insights of our work apply more broadly. For example, imagine the distribution over covariates $D_X$ is not fixed but also depends on $f$. Then, the problem of identifying $h^*$ is further complicated because we also need to establish overlap on $X$. However, once overlap is established our assumption enables identifiability. Thus, the challenges of $X$ and $\hat Y$ being coupled, the insights that incongruence can help tackle this, and the necessity of modeling $\hat Y$ explicitly to learn transferable models remains. Thus, we believe it is an important dimension of performativity that brings forward unique challenges that merit investigation.
>
> **Related work.** We thank the reviewer for drawing our attention to related works. In particular, the line of work by Swaminathan and Joachims is interesting because they also emphasize the importance of logging information about policies to be able to do offline evaluation. In their case it is propensities that enable the mitigation of selection bias. In our case it is predictions that allow us to model concept shifts caused by predictions.
>
> However, in general, works on mitigating covariate shift by modeling test-to-training density ratios are complementary to our work. Only once identifiability through overlap is established and propensity scores are well defined, can these methods be applied. However, our focus is on understanding when the problem is tractable. Complementarily, we show that identifiability can be established through incongruence even in settings where propensity score weighting is not applicable.
>
> The line of work on algorithmic recourse focuses on the relationship between covariates and predictions whereas we focus on down-stream effects of predictions on outcomes. Naturally, for learning which covariates would have led to a desired prediction (which is a counterfactual quantity) they need some randomness in the covariates. In our case it is useful to have randomness in the prediction for a fixed set of covariates because we want to understand the effect of the prediction on Y. This is a different problem that brings its own technical challenges. We will make this distinction more explicit in the related work.
>
> **Do the results provide a complete picture?** Our causal identifiability results pinpoint interesting properties of prediction functions that are sufficient to render causal effect estimation of predictions possible. The ability to separate the causal effect of $X$ and $\hat Y$ on $Y$ is crucial for identifiability without overlap. There might be other special configurations/assumptions that achieve the required degree of separability, but it seems difficult to provide a complete picture of these assumptions.
>
> **Quantify necessity of observing $\hat Y$.** Our causal model in Figure 1 elucidates that the distribution $P(X,Y)$ in the presence of performativity depends on $f$. Without observing $\hat Y$, is it impossible to estimate $E[Y|do(\hat Y), X]$, like without observing the treatment, one can estimate the effect of the treatment. By showing that without knowing $\hat Y$ there is an inevitable extrapolation gap that one suffers (Proposition 1) we show that this infeasibility result is relevant for prediction. In turn if one is able to observe $\hat Y$ we can hope to recover the causal effect of predictions to mitigate the extrapolation gap. In this sense, we see Proposition 3-6 as examples of positive results that are only possible because we had access to the predictions. In these settings the extrapolation gap between different models directly reflects the penalty for not having access to this information, quantifying the value of recording $\hat Y$.
>
> We hope this discussion clarifies your questions and the difference between the effect we study in our causal model and the type of distribution shifts studied in strategic classification or policy evaluation problems which thus bring forward different technical challenges.
>
> Please do not hesitate to follow up if you feel some questions remain unanswered.

---

> > ### Author Response · Authors · 2022-08-08
> > **Taking advantage of discussion phase**
> >
> > Dear Reviewer rWDi,
> >
> > The discussion phase is ending soon and we will not be able to respond to your comments after that.
> >
> > We would appreciate if you could let us know whether our reply to your review (including the general remark) and the discussion of related work addressed your concerns. If there is anything else you think we should do to further improve the paper, please let us know.
> >
> > Thank you!

---

> > ### Comment · Reviewer_rWDi · 2022-08-08
> > **Reviewer response**
> >
> > Thanks for the detailed response and the insightful comments. In order to achieve positive impacts on the community, discussion of limitations and alignment of expectations are crucial. As long as limitations are properly discussed, I agree with your position on causal models, related work, and completeness.
> >
> > Additionally, I would like to thank you for your detailed response to the question of "quantifying the necessity of observing $\hat{Y}$". Logging prediction data seems like a good practice in general. However, using predictions as a feature may raise concerns among practitioners due to the introduction of new dependency structures and feedback loops within the system. Although it may be the case that using predictions as features leads to positive outcomes when the causal structure is as shown in Figure 1, I am not sure whether it will always result in better results. In this regard, I am wondering whether the suggestion of incorporating predictions as features can be quantified in a way that is robust to the actual causal structure in some sense, and whether there are cases in which such a practice would be inappropriate.
> >
> > In light of your concrete suggestions for improvement and your discussion of limitations, I have raised my rating to "Borderline Reject". It is slightly below the threshold for acceptance due to the fact that the changes, while clearly outlined, appear to be extensive, and will not be peer-reviewed if the paper gets accepted. In order to ensure that this formal rating is appropriate for this scenario, I will consult with the area chair and adjust accordingly. Once again, thank you for your comments! I am looking forward to seeing the eventual outcome of this work.

---

> > > ### Author Response · Authors · 2022-08-09
> > > **follow-up**
> > >
> > > Thank you for your constructive feedback. We understand your concern and we make sure any potential for misunderstanding is eliminated.
> > >
> > > More specifically, we will do the following to be more forthright about the limitations of our assumptions:
> > > - adjust the wording in abstract and introduction to more closely tie the data-driven strategy 'predicting from prediction' to the identifiability assumptions.
> > > - add a dedicated *limitation section* to discuss our assumptions together with potential violations (other ways $f_\theta$ can be performative, anti-causal prediction, spill-over effects).
> > > - include a sentence along the lines of: *our results do not justify the use of supervised learning to address performativity in broader generality beyond the scope of our identifiability results.*
> > > - add more discussion of Figure 4(a)
> > >
> > > We would also like to note that the paper already makes an effort in being explicit about assumptions: The assumptions are explicitly stated in all our formal results and we never tried to hide them. We explicitly bring up strategic classification (L83-87) to make clear our causal model covers only a special case of performativity. We have a dedicated section (Section 4) to point to the limitations of the SUTVA assumption underlying our analysis. And in the empirical section we included a result  (Figure 4(a)) to demonstrate that supervised learning can go wrong if the identifiability assumptions are not satisfied. That said, we view these changes as an important improvement of existing efforts of being explicit about limitations, rather than an extensive change to the paper.
> > >
> > > We hope that this discussion and the more specific descriptions of the changes offer you more confidence in us that we have the ability to incorporate these changes properly in the paper.  We'd also be happy to incorporate any further feedback if there is something you would like to add.

---

### Official Review · Reviewer_e7sJ · 2022-07-12

**Rating:** 6
**Confidence:** 3
**Soundness:** 3 good
**Presentation:** 3 good
**Contribution:** 3 good

**Summary:**

The setting of the paper is about making predictions from predictions -- given a deployed model and decision subjects may best respond to it, how will their target variable Y change. The goal for the authors is to predict the impact of a new model deployment before actually deploying it. Instead of trying to come up with an optimal solution (e.g. performative optimality), the authors are interested in understanding the underlying causal mechanism of the distribution shifts.

**Questions:**

Can the author provide some insights on why overlapping is easier to achieve if $|\mathcal{T}|$ is small? In general, can we boost the overlapping by including more classifiers? If the underlying causal model is more complicated (e.g. have more variables $X_1, X_2..$ or more complicated causal relationships among each other), what will be a good way to ensure the identifiability of the causal structure?

**Limitations:**

Please see the strength and weaknesses section.

**Strengths And Weaknesses:**

I find the problem of identifying $E[Y| do (\hat{Y} = y), X]$ in the performative prediction setting interesting. Overall the paper makes a novel contribution by providing sufficient conditions to identify the causal effect of predictions under some assumptions. The relation to the literature on spillover effect/social network analysis is also interesting.

To me, the major weakness to me is that the paper seems to make heavy assumptions in order to have interesting and clean results. For example, the whole paper is built on assuming a particular causal structure (Figure 1),  and the definition of the extrapolation error allows a clean separation between the influence of $X$ and $\hat{Y}$, and in section 3.3, the author assumes the effect of $X$ on $Y$ and $\hat{Y}$ is separable as well. Thus it wasn't clear to me how likely this work can be extended to more complicated settings. The authors provide some empirical justification, but I would like to see some theoretical insights on how the results would change/not hold without certain assumptions, and I believe it will greatly strengthen the paper.

---

> ### Author Response · Authors · 2022-08-02
> **Author response**
>
> **Overlap.** By stating that overlap is easier to satisfy we mean that for smaller set $|T|$ it is more reasonable to assume positive probability for every event. But this should not be interpreted as a formal claim, we can remove it if it is confusing.
>
> **Multiple environments.** This is an interesting point. If we have data from the deployment of multiple models (corresponding to different distributions) we can pool the data to increase overlap. However, for deterministic prediction functions we would still need an unbounded number of environments to achieve global identifiability. The reason is that for every $X$ every single model only provides an estimate for one pair $(X,\hat Y)$. So it does not make the problem significantly less complex unless we are willing to make parametric assumptions.
>
> **More complex causal structures.** Unfortunately, causal inference on observational data is impossible without uncheckable causal assumptions on how the outcome is generated (Pearl, 2009). Thus it is unavoidable to focus on one particular graph. By studying the graph in Figure 1 we want to draw attention to the causal link between the prediction $\hat Y$ and the outcome $Y$ and pinpoint the challenges of learning this causal effect due to the coupling between $X$ and $\hat Y$. This is a general challenge that persists also in the presence of additional performative effects.
>
> We see two valuable extensions of our graph:
> - First, the graph could be extended to allow $f$ to impact the distribution over covariates $D_X$ and thus embrace a broader class of performative effects (e.g., strategic behavior). In this case we would, in addition, need to deal with overlap in $X$. But, once overlap in $X$ is established, our assumptions would still ensure identifiability.
> - A second extension would be a graph where not all features in $X$ are necessarily causal for $Y$. However, if $\hat Y$ induces changes in $Y$ that propagate to these anti-causal features this leads to cycles in the causal graph that demand additional caution and violate our assumptions. Prior work has demonstrated that anti-causal prediction should be avoided in the presence of strategic manipulations, which supports our idealistic assumption of causal prediction.  However, such features might nevertheless be used for prediction, and an investigation of performativity in the context of anti-causal predictions would be an interesting direction for future work.
>
> **On the role of separability.** Since overlap is not satisfied for deterministic prediction functions we wish to see what is possible beyond assuming overlap. The challenge is that without overlap we need to predict unobserved counterfactual quantities. To achieve this we need to be able to disentangle the direct from the indirect link in order to extrapolate. Without some sort of separability in $g(X, \hat Y)$, the function $g(.)$ and the interventional quantity $E[Y|X, do(\hat Y)]$ is not identifiable: there exists $g’(.)$ different from $g(.)$ such that $g’(X, \hat Y) = g(X, \hat Y)$ under $\hat Y = f_\theta(X)$. One such example is the function $g’(.)$ that satisfies $g’(X, \hat Y) = g’(X) \stackrel{def}{=}  g(X, f_\theta(X))$. Such a g’(.) may be compatible with the observational data as $g(.)$ does but could imply different values of the interventional quantity $E[Y|X, do(\hat Y)]$. That said, under separability, this issue of non-identifiable $g(.)$ does no longer exist, and can enable causal inference on $E[Y|X, do(\hat Y)]$.
>
> If only parts of $g$ are separable, then overparameterization will help us correctly identify these separable components. Similarly, if separability is violated discrete classification will allow for the effects to be identified locally but not globally (akin to an RDD).  In any case, weak violations of our assumptions no longer allow for conclusions about *any* model $f_\phi$, however they might still be useful to extrapolate to *some* models. To make explicit how violations of the assumptions shrink the set of models we can extrapolate to, we added a refined randomness result sketched in the general comment. From a practical perspective this implies that we should avoid drawing conclusions about models that are very different from $f_\theta$ if we can not be sufficiently confident that our assumptions hold.

---

> > ### Comment · Reviewer_e7sJ · 2022-08-07
> > **Thanks for the detailed responses!**
> >
> > I have carefully read the authors' responses, and my evaluation remains the same.

---

### Author Response · Authors · 2022-08-02
**General comment to all reviewers**

Thank you to the reviewers for their valuable feedback and appreciation of our work. Before we respond to the questions by the reviewers individually, we would like to clarify one concern upfront.

**Bigger picture.** We want to be clear that we are not claiming to ‘solve’ performative prediction or advocate for predicting from predictions as a sufficient solution to tackle performative distribution shifts in full generality. Our work focuses on one specific causal effect and we want our technical results to be understood as a proof of concept that shows how access to the predictions could enable positive results in the presence of performativity. We hope this contributes an additional perspective to recent discussions about data collection for machine learning, in particular emphasizing the importance of collecting information about predictions. We highlight this as a necessary step to enable fruitful research that builds better understanding of the down-stream consequences of predictions.

Beyond this broader goal, we believe our causal identifiability results are of independent interest because they distill interesting challenges and particularities of performing causal inference when the treatment variable is not randomly assigned, but based on a prediction output by a machine learning model.

---

> ### Author Response · Authors · 2022-08-02
> **How we plan to make use of the additional content page**
>
>
> In response to the reviews we will make use of the additional content page as follows:
>
> 1. We will add a discussion section to make the *limitations* of our assumptions clear and explain that our results do not justify predicting from predictions as a general purpose strategy for anticipating performativity. It is important to us that this is not misunderstood to ensure that this work has a positive impact on the community in the long run.
> 2. We will expand the related work to discuss the papers pointed out by the reviewers and do some further literature search tracing the references therein.
> 3. We will anticipate questions and add clarifications where needed.
> 4. We will add a refined randomness result to Section 3.2 that makes explicit the set of models we can extrapolate to even if global identifiability can not be established. More formally; we can extrapolate to models $f_\phi$  for which $\forall x\in\mathcal X$ and subsets $\mathcal Y'\subseteq\mathcal Y$ with positive measure it holds that $P[f_{\phi}(x)\in \mathcal Y']/P[f_\theta(x)\in \mathcal Y']>0$.
> This assumption puts our result apart from classical overlap assumptions in that it takes into account that $\hat Y$ is the output of a known prediction function, and the target ‘environment’ is fully specified by $f_\phi$.

---

### Meta-Review · Area_Chair_EpNw · 2022-08-24

**Recommendation:** Accept
**Confidence:** Certain

**Metareview:**

Strengths:
* problem is well motivated and stated, writing is clear
* interesting and important identification problem
* useful results on sufficient conditions
* causal assumptions are made explicit (see also weaknesses)

Weaknesses:
* very strict assumptions on causal structure (specifically that $\hat{Y}$ does not affect $X$)
* assumptions made explicit, but writing conveys more general claims at the onset
* theoretical results make interesting connections but lack clear novelty
* current version missing a thorough discussion of limitations
* concerns regarding using predictions as a feature

Summary:
The paper presents an interesting study of the effects of predictions on outcomes. The key contributions are a clean formulation of the problem, several identifiability results, and some corollaries regarding the use of predictions as input for learning.
Reviewers were unanimous in their appreciation of the paper’s quality—but also in their concern regarding the strong assumptions made. While making assumptions is certainly adequate, some reviewers felt that this narrows the scope of how results should be interpreted. One reviewer was also worried that assumptions simplify the problem in a way that makes the paper’s theoretical contributions derive immediately from know results in causal inference. But the main concern was that, while assumptions were indeed clearly stated, earlier sections seem to present the paper as being more general than it is, thus creating false expectations and deferring a much needed discussion regarding the paper’s limitations that follow from its strong assumptions.
In the discussion, reviewers were mostly satisfied with the authors’ responses as to how they plan to address the concerns raised. Unfortunately, the authors have only stated what changes they intend to apply, and did not provide reviewers with a revised version. This makes judging these anticipated changes difficult. Nonetheless, all reviewers consider the paper and its contributions favorably; the authors are therefore strongly urged to clearly and adequately frame their paper’s results and limitations, with full integrity, and as early in the paper as possible.


**Award:**

No

---

### Decision · Program_Chairs · 2022-09-14

Accept